# Exploring differences between gender expressions in exercise self-efficacy and outcome expectations for exercise in individuals with stroke

Elise Wiley[1], Kevin Moncion[1], Lynden Rodrigues[2,3], Hanna Fang[1], Kenneth S. Noguchi[1], Marc Roig[2,3], Julie Richardson[1], Joy C. MacDermid[1,4], Ada Tang[1] *

1 School of Rehabilitation Science, McMaster University, Hamilton, Ontario, Canada, 2 Feil and Oberfeld Research Centre, Jewish Rehabilitation Hospital, Laval, Quebec, Canada, 3 School of Physical & Occupational Therapy, McGill University, Montreal, Quebec, Canada, 4 School of Physical Therapy, Western University, London, Ontario, Canada

* atang@mcmaster.ca

**Data Availability Statement:** Data are available through the McMaster University Dataverse via

## Abstract

Gender expression may be associated with exercise self-efficacy and outcome expectations for exercise in the general population. Exercising for challenge and enjoyment are associated with the instrumental traits typically held by individuals with masculine gender expressions. Conversely, exercising for weight loss to receive validation from others are in line with the expressive traits most commonly held by individuals with feminine gender expressions. Moreover, possessing neither dominant nor expressive traits (undifferentiated gender expressions) have been linked to poorer psychological outcomes. Exercise is important after stroke, but gender differences in psychosocial factors for exercise in this population were unknown. The purpose of this study was to explore whether gender expression differences exist in exercise self-efficacy and outcome expectations for exercise post-stroke. Gender expression (masculine, feminine, androgynous, undifferentiated) was assessed using the Bem Sex-Role Inventory-12 (BSRI-12) in 67 individuals with stroke. Self-efficacy and outcomes expectations for exercise were assessed using the Self-Efficacy for Physical Activity Scale and Short Outcome Expectations for Exercise Scale, respectively. One-way analysis of covariance models were conducted, adjusting for biological sex, age, and time post-stroke. There were differences in exercise self-efficacy across the four gender expression groups (F(3,60) = 4.28, p<0.01), where individuals with masculine gender expressions had higher self-efficacy than those with undifferentiated gender expressions (adjusted mean: 3.56 [SE: 0.17] vs. 2.72 [SE:0.18], p<0.01). There were no differences in outcome expectations for exercise (F(3,57) = 1.08, p = 0.36) between gender expressions. In our pairwise comparisons, we found that individuals with masculine gender expressions had higher exercise self-efficacy than individuals possessing undifferentiated gender expressions. Strategies to enhance exercise self-efficacy after stroke are needed, particularly for individuals with undifferentiated gender expression. There were no associations between gender expression and outcome expectations for

https://doi.org/10.5683/SP3/T3Y8P1 The longer format can also be found here: https://borealisdata.ca/dataset.xhtml?persistentId=doi:10.5683/SP3/T3Y8P1.

**Funding:** The baseline data collected from the randomized controlled trial (HIREB 4713 & CRIR-1310-0218) was supported by an operational grant from the Canadian Institutes of Health Research (388320). The Canadian Institutes of Health Research had no role in the design of this study, its execution, analyses, interpretation of the data, or decision to submit results.

**Competing interests:** The authors have declared that no competing interests exist.

exercise after stroke. Clinicians may continue reinforcing the positive expectations towards exercise across all gender expressions.

## Introduction

Exercise is known to have many benefits for individuals living with stroke, with strong evidence supporting improved cardiovascular fitness, walking function, balance [1], cognitive function [2], and reductions in vascular risk factors, including cholesterol and blood pressure levels [3, 4]. Despite the known benefits and best practice recommendations to engage in regular exercise [5], physical activity levels after stroke are substantially lower than older adults without stroke [6]. Individuals with stroke also report low levels of exercise self-efficacy [7–10] and lower expectations on the benefits of exercise [11–14], which can influence the initiation and adherence of exercise [15]. Moreover, stroke physiotherapists have also acknowledged that exercise training is not typically prioritized within management plans or as part of stroke rehabilitation goals [16, 17], which may indeed further limit opportunities for positive exercise behaviours.

There have been preliminary reports of gender-based considerations being associated with participation in stroke rehabilitation programs, specifically around barriers faced by women. With regards to the construct of gender roles (representing the behavioral norms applied to men and women in society) [18], women with stroke are more likely to possess primary care-giver roles, whereby it is increasingly difficult to focus on their own health when pressures to prioritize the needs of others are present [19]. Moreover, women are less likely to view stroke as a major health concern which limits their motivation to participate in stroke rehabilitation [19]. These gender roles may also influence exercise behaviours.

Gender expression is a construct of gender that has not been studied in individuals with stroke but may be an important factor associated with exercise self-efficacy and outcome expectations for exercise. Gender expression refers to the traits, thoughts and beliefs an individual possesses regarding their gender self-concept, whereby gender expression is most commonly expressed on a fluid spectrum of masculinity to femininity [20]. Importantly, an individual's gender expression may differ from their biological sex at birth [18]. Sex is a biological construct, whereby an individual is characterized as being male or female according to genetics, anatomy, and physiology [18], which influence biological processes such as ageing and prevalence, diagnosis, severity, and outcomes of disease [21].

Sandra Bem was the first to propose that individuals with masculine gender expressions were more likely to take on instrumental traits, whereas individuals with feminine gender expressions were more likely to encompass expressive traits [22]. Dr. Bem also proposed two additional cross-sex-typed genders: androgynous (possessing both masculine and feminine traits) and undifferentiated (below-average propensities of both masculinity and femininity) [22].

An individual's gender expression may have an important influence on their exercise self-efficacy and outcome expectations for exercise, whereby differences in exercise behaviours between individuals possessing masculine versus feminine gender expressions are reported early in life and persist into adulthood. Specifically, people with masculine gender traits possess higher exercise self-efficacy than those with feminine gender traits, and thus are more likely to participate in exercise [23–31]. Men are also more likely to report challenge, enjoyment, and gains in strength and fitness as exercise motives whereas women report exercising for stress reduction or weight management to improve physical appearance [32–37]. Similarly, men are

also more likely than women to report positive outcome expectations for exercise, including greater enjoyment and improved satisfaction with leisure activities and quality of life from participating in exercise [28, 34, 38–40].

Factors associated with exercise behaviours in men, such as exercising for challenge and enjoyment, are associated with masculine gender expressions, dominated by instrumental traits encompassing personal agency and accomplishment [22]. Conversely, the motives for exercise in women, such as exercising for weight loss to receive validation from others, are in line with feminine gender expressions where expressive personality traits involving the maintenance of social relationships and focusing on the needs of others are present [22]. Indeed, previous research has supported that younger adults with feminine gender expressions possessed less favourable perceptions of an ideal body weight than those with masculine gender expressions [41]. Undifferentiated individuals have also been show to possess lower mental health and self-esteem compared to other gender expressions [42], while individuals possessing androgynous gender expressions may be positioned to possess the most favourable psychological wellness and health behaviours due to the optimal balance between instrumental and expressive traits [22, 41].

There is currently a gap in knowledge of whether gender-related associations exist in the context of psychosocial factors for exercise among individuals with stroke.

Therefore, the objective of this study was to determine whether gender expression (masculine, feminine, androgynous and undifferentiated) differences exist in exercise self-efficacy and outcome expectations for exercise among individuals with stroke. Based on previous literature, we hypothesized that either individuals with masculine or androgynous gender expressions would possess the highest exercise self-efficacy and outcome expectations for exercise, while individuals with feminine and undifferentiated expressions would demonstrate the least favourable self-efficacy and outcome expectations for exercise. Findings from this research may provide important insight for clinicians and researchers on the associations between gender-based factors with exercise behaviours among individuals with stroke, and thus can promote opportunities to deliver satisfactory interventions and treatments targeted to the specific traits of individuals with different gender expressions.

## Materials and methods

This study was a cross-sectional analysis of data collected from three studies: baseline data from a multi-site randomized controlled exercise trial [43] (Hamilton Integrated Research Ethics Board [HIREB] 4713, McGill University Centre de Recherche Interdisciplinaire en Réadaptation du Montréal Métropolitain CRIR-1310-0218) (n = 59; recruitment period April 1, 2019 –ongoing), and two prospective single-group studies (HIREB 3113 (n = 2; recruitment period February 26, 2018 –December 31, 2019) and 12734 (n = 6; recruitment period June 24, 2021- October 17, 2022)). All study procedures were followed in accordance to guidelines outlined by the respective institutional research ethics committees. Informed written consent was obtained from all participants. A member of the research team provided detailed explanations of the study logistics and responded to any questions, prior to the participant completing the consent form. The same study team member observed the participant complete the form (a trusted family member or friend was present, if needed). We used the STROBE cross-sectional checklist when writing our report [44].

### Participants

The study sample was one of convenience. Eligibility and recruitment strategies were similar for all three studies. Participants were recruited from the community, through local

community stroke groups and from a database of former research participants who consented to be contacted for future research studies. Members of the research team attended monthly meetings of community stroke groups and provided a brief presentation of the research study, distributed study flyers, addressed questions, and collected phone numbers of interested individuals. For all recruitment methods, participants were screened for eligibility for the study over the phone with a member of the study team. Individuals were eligible to participate if they were between 40–80 years old, at least 6-months post-stroke, living in the community and able to walk at least 10 meters independently (gait aids permitted). Individuals were excluded if they had a stroke of non-cardiogenic origin or tumor, scored >2 on the Modified Rankin Scale, or, relevant for the exercise studies, had any contraindications to exercise testing [45] or class C or D American Heart Association Risk Criteria. Individuals were also excluded if they presented with other neurological or musculoskeletal comorbidities, pain worsened with exercise, or cognitive, communication, or behavioural issues that could limit their ability to provide consent or follow instructions.

### Assessments

**Participant characteristics.** Participant demographic information, stroke lesion type and location, time post-stroke (years), degree of disability using the Modified Rankin Scale [46] were collected.

**Gender expression (independent variable).** The Bem Sex-Role Inventory-12 (BSRI-12), a 12-item questionnaire [47, 48] derived from the original 60-item BSRI questionnaire [22], was used to assess gender expression, as recommended by experts in sex and gender-based considerations [18]. The BSRI-12 consists of two scales with 6 stereotypical feminine gender traits (warm, gentle, affectionate, sympathetic, sensitive to other's needs, tender) and 6 stereotypical masculine gender traits (has leadership abilities, strong personality, acts as leader, dominant, defends own beliefs, makes decisions easily) [47, 48]. Participants rated the extent to which each trait reflected themselves on a 7-point Likert scale (1 = never or almost never true, 7 = always or almost always true) [47, 48].

We used the median-split to classify the BSRI-12 [49, 50]. This commonly used approach in older adult populations [51, 52] was selected as the alternative approach to using normative values which was based on a sample of younger adults [22], and thus not representative of our sample. The median-split method involved first determining the overall median value for both feminine and masculine subscales from the sample, then comparing individual participant scores from the masculine and feminine subscales [47]. Gender expression was classified for each participant as follows: feminine gender expression if the mean score on the feminine scale was higher than the overall median and mean score on the masculine scale was lower, masculine gender if the mean score on the masculine scale was higher than the overall median and mean score on the feminine scale was lower, undifferentiated if mean scores for both the feminine and masculine scales fell below the median, and androgynous if mean scores on both scales were equal to or above the median for feminine and masculine subscales [47]. The BSRI-12 has high internal consistency in older adults (feminine scale Cronbach's alpha 0.76, masculine scale Cronbach's alpha 0.75) and discriminant validity between the two separate masculine and feminine scales [53]. We also calculated the internal consistency of each subscale within our study sample.

**Psychosocial outcomes for exercise (dependent variables).** Albert Bandura's Social Cognitive Theory outlines a core set of psychosocial determinants of health that are required to successfully execute health behaviours [54, 55], such as exercise. Two of these core

determinants, self-efficacy and outcome expectations, were the primary dependent variables of interest in our current study.

*Self-efficacy for exercise.* Exercise self-efficacy is formally defined as a person's beliefs in their capabilities to successfully engage in bouts of exercise [56, 57]. Regular engagement in exercise among older adults promotes greater mastery experiences which can translate into psychosocial benefits such as reduced stress and depression [58]. There is a strong positive association between higher exercise self-efficacy and greater levels of physical activity engagement in older adults [59–61] and individuals with stroke [62].

With the 5-item Self-Efficacy for Physical Activity Scale (SEPA) [63], participants were asked to rate their levels of confidence in their abilities to exercise using a 5-point Likert scale ranging from 1 (not confident at all) to 5 (extremely confident). Scores obtained on each of item of the questionnaire were summed to calculate a mean score [63]. The SEPA has demonstrated strong internal consistency (Cronbach's alpha 0.76–0.85) and test-retest reliability ($r = 0.90$) in adults [63], and predictive validity for participation in physical activity guidelines in adults [64].

*Outcome expectations for exercise.* Outcome expectations for exercise reflects the belief that engaging in exercise behaviours will produce a specific outcome [65], whereby a positive association exists between higher outcome expectations and greater physical activity levels in older adults [61] and individuals with stroke [62].

The Short Outcome Expectations for Exercise Scale (SOEE) [66, 67] is a 5-item questionnaire that is used to evaluate the outcome expectations related to exercise that are relevant to older adults. Participants were asked to rate their expectation of positive outcomes for exercise on a 5-point Likert scale ranging from 1 (strongly disagree) to 5 (strongly agree), and scores obtained on each of item of the questionnaire were summed to calculate a mean score [67]. In individuals with stroke, the SOEE demonstrated both high internal consistency (Cronbach's alpha 0.90) and construct validity with common outcome expectations related to exercise, which include exercising to improve mood, alertness, endurance and believing that exercise is enjoyable ($\lambda = 0.67$–0.88) [67].

Cronbach's alpha and McDonalds' omega were computed to determine internal consistency of the SOEE and SSEE in our sample. To minimize the risk of bias, a member of the study team was present and assisted each participant by reading out the instructors and question/items within each questionnaire.

## Statistical analyses

Participant demographics were described using descriptive statistics for mean and standard deviations for normally distributed continuous variables, frequency (percentages) for categorical variables, and median and interquartile range for non-normally distributed data. Data were inspected for normality through histograms and the Shapiro-Wilk test. Assumptions for homogeneity of the variances were tested using the Bartlett's and Cooks-Weisberg Tests and distribution of the residuals were inspected using a skewness/kurtosis test and Quantile-Quantile (Q-Q) plots. In the SOEE models, three outliers with high residuals and influence were removed and models were re-evaluated.

We first conducted a one-way analysis of variance to determine if there were any differences between the four constructs of gender expression and psychosocial factors for exercise. Next, one-way analysis of covariance models were conducted, where we adjusted for biological sex [68], age [68], and time post-stroke [10, 69] due to the known associations with exercise behaviours post-stroke. Pearson correlation analyses were conducted to ensure that the

selected covariates were not highly associated. We also tested for an interaction between our independent variable (BSRI-12) and each covariate.

If the unadjusted and adjusted models indicated differences in psychosocial factors for exercise outcomes between BSRI-12 groups, we subsequently conducted Sidak-adjusted pairwise comparisons to explore where the differences between the four BSRI-12 groups lay. The contrast in adjusted means, accompanying standard errors, 95% confidence intervals, and p-values were reported for BSRI-12 groups with significant differences in SEPA and SOEE scores, respectively. The accepted significance level was set *a priori* to a $p<0.05$, and all statistical analyses were performed on Stata/IC 15.1 (StataCorp, College Station, TX, USA).

## Results

Data from 67 individuals (n = 43 males, 24 females) were included in this study. The supporting data are available on the McMaster University Dataverse (https://doi.org/10.5683/SP3/T3Y8P1). Participant demographics for the full sample and disaggregated by gender expression group are shown in Table 1. The sample median value was 6 and 5.3 on the feminine and masculine subscales of the BSRI-12, respectively. The masculine (Cronbach's alpha 0.86, McDonald's Omega 0.87) and feminine subscales (Cronbach's alpha 0.89, McDonald's Omega 0.89) of our sample had high internal consistency.

The covariates were deemed to not be highly correlated ($r<0.13$), and were maintained in the models as there were no interactions observed with the BSRI-12 in either the exercise self-efficacy or outcome expectations for exercise analyses. There were no missing data for any of the independent or dependent variables of interest for the 67 participants included in the study, and there was a balanced proportion of participants in each of the gender groups of the BSRI-12.

**Table 1. Participant demographics for full sample (n = 67) and disaggregated by gender expression group.**

| | All | Masculine | Feminine | Androgynous | Undifferentiated | p-value |
|---|---|---|---|---|---|---|
| | N = 67 | N = 17 (25.4) | N = 17 (25.4) | N = 17 (25.4) | N = 16 (23.8) | |
| Sex, n (%) | | | | | | |
| Female | 24 (35.8) | 5 (29.4) | 11 (64.7) | 3 (17.6) | 5 (31.3) | 0.03 |
| Male | 43 (64.2) | 12 (70.6) | 6 (35.3) | 14 (82.4) | 11 (68.8) | |
| Age (years), Mean (SD) | 65.1 (8.6) | 63 (4.7) | 66 (7.1) | 65.1 (8.6) | 65.9 (10.6) | 0.81 |
| Years post-stroke, Median [IQR] | 2.1 [2.6] | 1.88 [1.7] | 2.43 [2.1] | 1.93 [2.8] | 2.3 [2.7] | 0.98 |
| Limb Affected, n (%) | | | | | | 0.12 |
| Right | 32 (47.8) | 8 (47.1) | 4 (23.5) | 11 (64.7) | 9 (56.2) | |
| Left | 34 (50.7) | 9 (52.9) | 13 (76.5) | 6 (35.3) | 6 (37.5) | |
| Bilateral | 1 (1.5) | 0 (0) | 0 (0) | 0 (0) | 1 (6.3) | |
| Stroke type, n (%) | | | | | | 0.69 |
| Ischemic | 46 (68.7) | 14 (82.3) | 10 (58.8) | 10 (58.8) | 12 (75) | |
| Hemorrhagic | 17 (25.4) | 2 (11.8) | 6 (35.3) | 6 (35.3) | 3 (18.7) | |
| Unknown | 4 (5.9) | 1 (5.9) | 1 (5.9) | 1 (5.9) | 1 (6.3) | |
| Montreal Cognitive Assessment, Median [IQR] | 26.0 [6.0] | 25.5 [2.5] | 28 [7.5] | 25 [12.0] | 27 [9.0] | 0.84 |
| Modified Rankin Scale, Median [IQR] | 1.0 [1.0] | 1.0 [1.0] | 1 [0] | 1 [2] | 2 [1] | 0.40 |
| Self-Efficacy for Physical Activity, Mean (SD) | 3.17 (0.78) | 3.6 (0.7) | 3.2 (0.8) | 3.1 (0.8) | 2.7 (0.6) | 0.01 |
| Short Outcome Expectations for Exercise, Mean (SD) | 4.13 (0.72) | 3.9 (0.7) | 4.3 (0.6) | 4.3 (0.7) | 4.0 (0.9) | 0.42 |

*Note.* IQR = Interquartile Range, SD = Standard Deviation.

**Table 2. ANCOVA of the association between self-efficacy for physical activity scores and the Bem Sex-Role Inventory-12 (n = 67 participants).**

| Variables | Partial Sum-of-squares | Degrees of Freedom | Mean Squares | *Root mean-square error = 0.71* | *R-squared = 0.25* |
|---|---|---|---|---|---|
| | | | | F-Statistic | P-value |
| Model | 9.89 | 6 | 1.64 | 3.29 | **<0.01**\* |
| BSRI-12 | 6.43 | 3 | 2.14 | 4.28 | **<0.01**\* |
| *Covariate*: Sex | 2.05 | 1 | 2.05 | 4.08 | 0.05 |
| *Covariate*: Time post-stroke | 1.76 | 1 | 1.76 | 3.51 | 0.07 |
| *Covariate*: Age | 0.00004 | 1 | 0.00004 | 0.00 | 0.9 |
| Residual | 30.08 | 60 | 0.50 | - | - |
| Total | 39.97 | 66 | 0.61 | - | - |

Note.

\*p<0.05

The SEPA demonstrated good internal consistency in our sample (Cronbach's alpha 0.75, McDonald's omega 0.76). The SEPA scores were normally distributed and there were no outliers. There were differences between gender expression groups observed [$F_{(3,60)} = 3.60$, p = 0.02]. Sidak corrected post-hoc analyses indicated that differences in unadjusted means were observed between masculine and undifferentiated gender groups [unadjusted mean: 3.59 (SD: 0.68) vs. 2.75 (SD: 0.64), p = 0.01]. Differences between the four gender constructs in exercise self-efficacy remained after controlling for age, sex, and time post-stroke [$F_{(3,60)} = 4.28$, p<0.01] (Table 2). Table 3 provides the adjusted means for each of the four gender groups. In pairwise comparisons, we found that individuals with masculine gender expressions had higher exercise self-efficacy than individuals with undifferentiated gender expressions [(adjusted mean: 3.56 (SE: 0.17) vs. 2.72 (SE:0.18), p<0.01)] (Fig 1 –Panel A; Table 3). There were no other differences in exercise self-efficacy between the four constructs of the BSRI-12 in either the unadjusted or adjusted models (Table 3).

The SOEE also demonstrated high internal consistency in our sample (Cronbach's alpha 0.86, McDonald's omega 0.87). Initially, the SOEE variable violated assumptions for normal

**Table 3. Mean SEPA scores between BSRI-12 groups adjusted for age, sex and, time post-stroke and Sidak correction adjusted post-hoc comparisons.**

| BSRI-12 | SEPA Scores | Standard Error | 95% CIs | |
|---|---|---|---|---|
| Feminine | 3.32 | 0.18 | 2.95, 3.69 | |
| Masculine | 3.56 | 0.17 | 3.21, 3.91 | |
| Androgynous | 3.04 | 0.18 | 2.69, 3.40 | |
| Undifferentiated | 2.72 | 0.18 | 2.36, 3.07 | |
| **BSRI-12 comparison** | **Contrast in Adjusted means** | **Standard Error** | **P-Value** | **95%CIs** |
| Undifferentiated verses Feminine\* | -0.60 | 0.26 | 0.12 | -1.30, 0.088 |
| Androgynous verses Feminine\* | -0.28 | 0.26 | 0.87 | -0.99, 0.43 |
| Masculine verses Feminine\* | 0.24 | 0.26 | 0.93 | -0.46, 0.93 |
| Undifferentiated verses Masculine\* | -0.84 | 0.24 | **<0.01**\*\* | -1.52, -0.17 |
| Androgynous verses Masculine\* | -0.52 | 0.24 | 0.21 | -1.18, 0.15 |
| Undifferentiated verses Androgynous\* | -0.32 | 0.25 | 0.72 | -1.00, 0.35 |

Note.

\*Denotes the reference group in the comparison. SEPA = Self-Efficacy for Physical Activity Scores, BSRI-12 = Bem Sex Role Inventory-12,

\*\*p<0.05

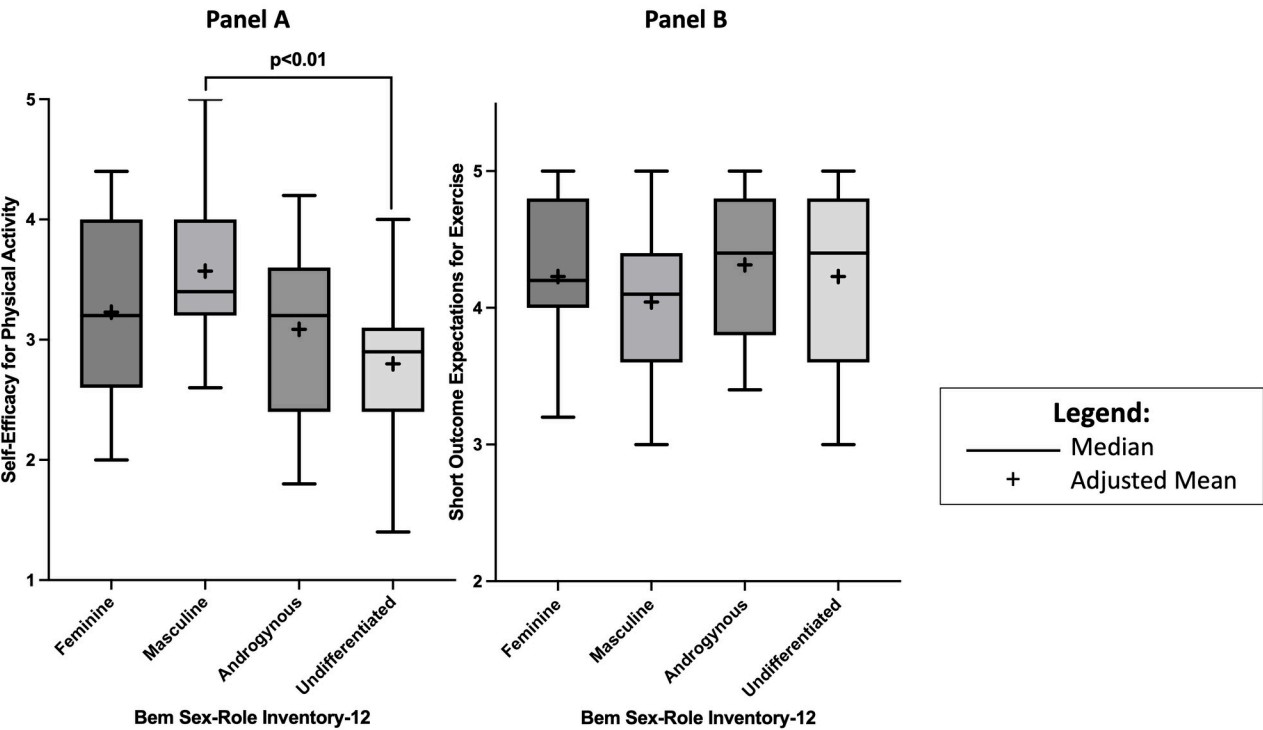

**Fig 1.** Box-and-Whiskers Plots of Self-Efficacy for Physical Activity scores (Panel A) and Short Outcome Expectations for Exercise (Panel B) for feminine, masculine, androgynous and undifferentiated gender expressions assessed by the Bem Sex-Role Inventory-12, adjusted for age, sex and time post-stroke.

distribution of residuals due to three major outliers observed in the Q-Q and fitted values against standardized residuals plots. All assumptions for distribution and homogeneity of variances were met, once the three outliers were removed. The outliers represent three different gender expression groups, ranged in age and predominantly experienced mild to moderate ischemic strokes. A detailed description of the characteristics of the three outliers are available as a supporting information file (S1 Table). There were no differences in unadjusted means between gender expression groups [F(3,60) = 0.88, p = 0.46]. Similarly, after adjusting for age, sex, and time post-stroke, no differences between the four constructs of gender were observed [F(3,57) = 1.08, p = 0.36] (Table 4 & Fig 1 –Panel B). No further post-hoc pairwise comparisons were conducted. Table 5 provides the adjusted means for each of the four gender groups.

## Discussion

This exploratory study is the first to contribute to the literature surrounding gender expressions differences in exercise self-efficacy and outcome expectations for exercise in individuals with stroke. Our findings suggest that individuals possessing masculine gender expressions had higher exercise self-efficacy than individuals with undifferentiated gender expressions. There were no differences among gender expressions and outcome expectations for exercise.

Our findings of higher exercise self-efficacy among individuals with masculine gender expressions are consistent with our hypothesis and previous literature in young [26, 27, 30] and older adults [23, 24, 28, 29, 31]. Clinicians and researchers may thus consider the assessment of gender through validated tools such as the BSRI-12 as standard practice in the delivery of stroke care. Masculine gender traits, such as possessing greater behavioural control (i.e.,

**Table 4. ANCOVA of the association between the Short Outcome Expectations for Exercise scores and the Bem Sex-Role Inventory-12 (n = 64 participants).**

| Variables | Partial Sum-of-squares | Degrees of Freedom | Mean Squares | Root mean-square error = 0.58 | R-squared = 0.14 |
|---|---|---|---|---|---|
| | | | | F-Statistic | P-value |
| Model | 3.14 | 6 | 0.52 | 1.59 | 0.17 |
| BSRI-12 | 1.07 | 3 | 0.36 | 1.08 | 0.36 |
| *Covariate*: Sex | 0.0021 | 1 | 0.0021 | 0.01 | 0.94 |
| *Covariate*: Time post-stroke | 2.00 | 1 | 2.00 | 6.06 | 0.02* |
| *Covariate*: Age | 0.26 | 1 | 0.26 | 0.80 | 0.38 |
| Residual | 18.82 | 57 | 0.33 | - | - |
| Total | 21.96 | 63 | 0.35 | - | - |

Note.

*$p < 0.05$

**Table 5. Mean SOEE scores between BSRI-12 groups adjusted for age, sex and, time post-stroke.**

| BSRI-12 | SOEE Scores | Standard Error | 95% Confidence Interval |
|---|---|---|---|
| Feminine | 4.31 | 0.14 | 4.03, 4.60 |
| Masculine | 4.04 | 0.14 | 3.74, 4.33 |
| Androgynous | 4.37 | 0.15 | 4.07, 4.68 |
| Undifferentiated | 4.17 | 0.15 | 3.86, 4.48 |

*Note*. SOEE = Short Outcome Expectations for Exercise Scores, BSRI-12 = Bem Sex Role Inventory-12

perceived ease of performing exercise), have shown to contribute positively to beliefs in successfully engaging in exercise [70]. In contrast, the likelihood of possessing traits favourable for exercise engagement may be lower in individuals with undifferentiated gender [42] (i.e. neither masculine or feminine gender expressions), thereby contributing to lower exercise-related mastery experiences that are primary drivers of exercise self-efficacy [71]. Moreover, individuals with undifferentiated gender traits are more likely to be at a greater risk of mobility disability [51] and report greater levels of social anxiety [72] and depression [73], in contrast to other gender expressions. Indeed, low self-efficacy is often a precursor for social anxiety [74]. The current study adds to the growing body of evidence of poorer health outcomes of individuals with undifferentiated gender traits, and highlights the need to develop and incorporate strategies to promote exercise self-efficacy after stroke that are specific to the needs of these individuals. Such strategies may include greater emphasis placed on individualized exercise programs with clinicians working 1:1 with patients, or involving other patients with similar functional abilities since individuals with undifferentiated gender expressions may not possess competitive nor dominant traits, nor focus on the perceptions and needs of others [22, 42]. These strategies may not only promote exercise self-efficacy but may also minimize social anxiety.

Outcome expectations for exercise did not differ between the four gender expression groups, which is comparable to literature in an older adult population (mean age 77 years old) [52] but differ from younger adults (range of mean age 18–48 years old) [34, 38–40]. It may be that factors facilitating exercise are different between younger and older adults. A study of older men and women across a range of gender expressions hypothesized that individuals with instrumental traits would possess the most favourable outcomes in self-reported physical health function, wellness, and life satisfaction but in fact found no differences between the four

groups [52]. In contrast, factors contributing to higher enjoyment and positive outcome expectations for exercise among younger individuals with masculine genders, include gains in fitness [34], higher self-rated health [40], and satisfaction with leisure activities [38, 39], which may not as relevant to older adults and those with stroke. Indeed, a systematic review of 6 qualitative and focus group studies of participants with stroke (n = 175, 54–71 years old) reported that the primary facilitators to positive exercise behaviours included social support from family members and clinicians [75], which may contribute to positive perceptions of exercise. The reported barriers were primarily related to environmental factors (e.g., transportation, costs) and physical impairments after stroke [75]. Only one study described psychosocial factors related to outcome expectations, such as the belief that exercise will not improve their condition and being boring and/or monotonous [14]. The positive outcomes expectations of exercise after stroke were related to enjoyment, improving mood and feeling better as reported by individuals across all genders in our study, thus reinforcing exercise behaviours as an important rehabilitation goal for individuals with stroke.

It is worth noting that the median mRS score (1 [IQR 1]) for the current sample would classify disability from stroke as mild, which may also have contributed to the relatively high outcome expectations for exercise. A recent study of 87 individuals with stroke reported that higher levels of disability was associated with lower engagement in physical activity [76]. We note however that nearly one-third of the sample presented with mRS scores greater than 3, and barriers to physical activity engagement included beliefs that physical activity was unsafe, fear of falling and causing pain, and being too tired [76]. Thus, level of disability could have a moderating role on the association between gender expression and perceptions of physical activity outcomes after stroke. Given that our sample encompassed individuals of relatively low disability measured by mRS, we were not able to explore this consideration further but suggest this may be an area of future research.

## Strengths

There remains a need for validated and comprehensive gender indices to better understand gender expression among individuals with stroke, however a strength of this study was the equal distribution of participants in each group of the BSRI-12, including individuals possessing androgynous and undifferentiated gender expressions. There has been limited research to date focused on gender expression differences in psychosocial outcomes after stroke and no evidence in cross-sex-typed genders. Our study, examining psychosocial outcomes for exercise across sex-typed and cross-sex-typed individuals with stroke offers a balanced perspective to the current body of literature. Further research may continue to build on this work with the goal of providing evidence to support targeted stroke rehabilitation to address the psychosocial needs of individuals with stroke across gender expressions.

## Limitations

This study has several limitations. Firstly, we recognize that the traits of masculinity and femininity captured by the BSRI-12 may not reflect those of today's society, but this is the most commonly used and recommended questionnaire [18] to capture the construct of gender expression. Moreover, our study did not take into account other psychological variables that may be associated with exercise behaviours, such as apathy and depression yet are common after stroke, especially among women [77, 78]. We also did not collect data on current physical activity levels or pre-stroke gender expression or exercise behaviours, and thus were unable to determine how the stroke further affected exercise self-efficacy across gender expressions. Future research may explore the fluidity of gender by exploring longitudinal analyses of the

associations of gender expression and exercise behaviours prior to and after stroke. Moreover, given the exploratory nature of this study and sparsity of previous literature on gender-based based considerations in this population, we did not perform a sample size calculation. Our data may inform future research to consult our group means and variances. Finally, authors of a previous study in older adults with arthritis have suggested that a ceiling effect on the SOEE questionnaire may exist, given the high scores obtained on the questionnaire (mean 4.1 out of 5) [79]. As such, future research may consider using other measures to explore outcome expectations for exercises in clinical populations.

## Conclusions

Results from this exploratory study are the first to provide insight into the associations between gender expressions and psychosocial variables may affect exercise engagement in individuals living with stroke. We observed that there are gender differences in exercise self-efficacy, where individuals possessing masculine gender expressions had higher exercise self-efficacy than individuals with undifferentiated gender expressions. Our findings also suggest that although psychosocial changes are common after stroke, gender expression may not influence outcome expectations for exercise to the extent that we may have expected. Overall, our findings highlight the need for strategies such as 1:1 patient-to-therapist ratio exercises programs or involving other individuals with similar functional abilities, to optimize exercise self-efficacy among cross-sex-typed individuals with stroke, and continued reinforcement of the positive perceptions and expectations towards exercise across all gender expressions.

## Supporting information

**S1 Table. Characteristics of the outliers removed from the model exploring the differences in gender expression and outcome expectations for exercise.**
(DOCX)

## Acknowledgments

The authors would like to acknowledge the participants that invested their time in participating in this research study.

## Author Contributions

**Conceptualization:** Elise Wiley, Julie Richardson, Joy C. MacDermid, Ada Tang.

**Data curation:** Elise Wiley, Kevin Moncion, Lynden Rodrigues, Hanna Fang, Kenneth S. Noguchi, Marc Roig, Ada Tang.

**Formal analysis:** Elise Wiley.

**Methodology:** Elise Wiley, Ada Tang.

**Project administration:** Ada Tang.

**Resources:** Kevin Moncion, Lynden Rodrigues, Hanna Fang, Kenneth S. Noguchi, Marc Roig.

**Software:** Elise Wiley.

**Supervision:** Ada Tang.

**Validation:** Elise Wiley.

**Visualization:** Elise Wiley.

**Writing – original draft:** Elise Wiley.

**Writing – review & editing:** Kevin Moncion, Lynden Rodrigues, Hanna Fang, Kenneth S. Noguchi, Marc Roig, Julie Richardson, Joy C. MacDermid, Ada Tang.

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
