## [Decision Letter · Decision Letter 0]

28 Nov 2023

PONE-D-23-30818Exploring gender differences in exercise self-efficacy and outcome expectations for exercise in individuals with strokePLOS ONE

Dear Dr. Tang,

Thank you for submitting your manuscript to PLOS ONE. After careful consideration, we feel that it has merit but does not fully meet PLOS ONE’s publication criteria as it currently stands. Therefore, we invite you to submit a revised version of the manuscript that addresses the points raised during the review process.

We look forward to receiving your revised manuscript.

Kind regards,

Henry Hugh Bailey, Ph.D

Academic Editor

PLOS ONE

Journal Requirements:

4. We note that you have stated that you will provide repository information for your data at acceptance. 

Should your manuscript be accepted for publication, we will hold it until you provide the relevant accession numbers or DOIs necessary to access your data. If you wish to make changes to your Data Availability statement, please describe these changes in your cover letter and we will update your Data Availability statement to reflect the information you provide.

5. Please include a caption for figure 1.

**Additional Editor Comments:**

Thank you for submitting your manuscript to PLOS ONE. Please address the issues raised by the reviewers- in particular, please carefully address the statistical issues raised by Reviewer #3.

Reviewers' comments:

Reviewer's Responses to Questions

**Comments to the Author**

1. Is the manuscript technically sound, and do the data support the conclusions?

Reviewer #1: Yes

Reviewer #2: Partly

Reviewer #3: Partly

2. Has the statistical analysis been performed appropriately and rigorously? 

Reviewer #1: Yes

Reviewer #2: No

Reviewer #3: Yes

3. Have the authors made all data underlying the findings in their manuscript fully available?

Reviewer #1: Yes

Reviewer #2: Yes

Reviewer #3: Yes

4. Is the manuscript presented in an intelligible fashion and written in standard English?

Reviewer #1: Yes

Reviewer #2: Yes

Reviewer #3: Yes

5. Review Comments to the Author

Reviewer #1: Thank you for the opportunity to review this well written manuscript. I have some minor suggestions to improve the manuscript before publication detailed below.

General:

Gender and gender expression are used interchangeably throughout. Please review the manuscript and take care to explicitly refer to gender expression whenever relevant.

Title: The title needs to clarify that this refers to gender expression and not gender.

Introduction: There needs to be better clarification of what gender expression is and how it differs from biological sex and gender identification e.g. I may be born female, identify as female but have a male gender expression. This differentiation was not clear to me, as a novice on gender expression, until I read the section on the BSRI-12.

Although paragraph 3 introduces gender expression a more detailed description of gender expression and its constructs is warranted.

Paragraph 4 of the introduction refers to gender differences in exercise and outcome. Please clarify if this refers to biological sex, self-identified gender or gender expression.

Results:

It would be of interest to see what masculine expressions are related to exercise self-efficacy and outcome expectation. Would the authors consider analysing these 12 criteria separately. This may change the recommendation for clinical practice to not test for gender expression as a whole but those specific criteria. Your discussion does also point to other research which has looked at these individual characteristics.

Discussion:

Please take care to be consistent in your reference to gender and gender expression.

I would also question whether left/right dominance and the side affected by the stroke may affect once perception of self-efficacy and exercise outcome. I would suggest that this is explored in the analysis or discussed as a consideration for future research/limitation.

Please check spelling of self-efficacy throughout as it is variable - hypenated, one word, two words.

Reviewer #2: Introduction:

The authors presented the introduction in a clear and structured manner. However, while the authors stated that there is a gap in knowledge about whether gender-related associations exist in the context of psychosocial factors for exercise, I feel that they need to expand a bit more on this in order to better explain the justification for the study and the study's original contribution to knowledge. The authors described the literature on stroke and exercise, and exercise and gender expression. I'm curious as to whether there is any literature about any gender-based factors in persons with stroke outside of exercise, which may add to the understanding of how gender-based factors may affect exercise in this population. This may help explain the 'gap' in the literature a bit better.

Analysis/Methods:

I have concerns about the sample size. There is no indication of a power analysis or sample size estimation calculations described so it is difficult to say whether the study was sufficiently powered.

Also, while the sample was one of convenience from community, and databases, how did they go about recruiting from these? What was their recruitment process - phone, flyers - how were participants contacted? How did they assess for the inclusion/exclusion criteria?

There were no tests performed on the independent variables to determine whether the groups were different in any of the demographic factors. Also, the gender groups and demographic characteristics of the outliers should be mentioned.

Is the median-split a valid way to categorise participants into gender groups? The authors gave no justification for doing the classification this way. How does the BSRI-12 classify persons? Why did the authors not use this method of classification? I did a brief search of the tool and found that the median-split has been used as a way to classify participants into gender roles. This should be better detailed in the study.

Overall:

There are a few grammatical errors but these do not affect the understanding of the article and can be easily rectified.

The suggestions for future research are very good given the cited limitations of the study.

Overall, the study is relevant to today's evolving society and highlights important considerations, however, there are some recommended corrections that I think would be helpful in improving the validity of the study.

Reviewer #3: I enjoyed reviewing this paper. The topic is quite interesting and relevant for tx planning in stroke rehab. The paper is generally solid in terms of clarity and style of writing. The analysis is articulated clearly.

I have indicated "major" revision somewhat arbitrarily, mainly because I have several comments that I think would strengthen the paper, but I think you will be readily able to address these.

First, a few straightforward editing recommendations:

Line 96: Word choice. Is "transpire" what you mean. Consider "persist" or "continue."

97: Should read: "...people with masculine gender traits are more likely..."

99: "reported" should read "report" (be consistent with tense)

116: should be "knowledge of" or "knowledge regarding"

124:The use of "outcomes" without qualification here is a bit confusing. Would be better to specify that you expect lower self-efficacy and expectations so the reader doesn't conflate scale scores with exercise outcomes.

137: "ethic" should be "ethics"

182-83: I suggest eliminating the text in parentheses.

300: "between" should be "among"

In general, you should be consistent with use of Oxford commas.

Now, a few more substantive recommendations:

1. When you first mention outliers, the number is not indicated. Later, you mention removing outliers in the analysis of the SOEE. I think you provide a clear statement about all outliers removed and when. This is especially important given your relatively small sample size.

2. I see that you reported alphas for the BSRI subscales, but I don't see them for the other instruments. Reporting internal consistency for all scales is a good idea. Also, you might consider reporting McDonald's Omega, given criticisms of the restrictive assumptions of alpha. I recently had this critique myself from a reviewer and just reported both.

3. You mention some literature on people with undifferentiated traits, but I think this deserves more elaboration in your discussion. You expected to find other differences as well between masculine and feminine traits, so the finding stands out. More discussion on personality correlates and health outcomes for people with undifferentiated traits would flesh out the paper--and be relevant for clinicians working with this group.

4. Your sample size mey have limited power to find some differences. Addressing this explicitly in limitations and recommendations would add to the paper.

6. PLOS authors have the option to publish the peer review history of their article (what does this mean?). If published, this will include your full peer review and any attached files.

Reviewer #1: No

Reviewer #2: No

Reviewer #3: **Yes: **Michael H. Campbell

---

## [Author Response · Author response to Decision Letter 0]

4 Jan 2024

Response Document: “Exploring gender differences in exercise self-efficacy and outcome expectations for exercise in individuals with stroke”

We thank the reviewer for their helpful comments to strengthen the manuscript even further. We have provided a point-by-point response to each comment in below, and have uploaded a clean and revised manuscript to the portal. Additions to the manuscript have been added in yellow highlight. 

Reviewer 1

Thank you for the opportunity to review this well written manuscript. I have some minor suggestions to improve the manuscript before publication detailed below.

General

Gender and gender expression are used interchangeably throughout. Please review the manuscript and take care to explicitly refer to gender expression whenever relevant.

AUTHOR RESPONSE: We thank Reviewer #1 for their comments and suggestions to improve the manuscript. We appreciate the attention to detail with this comment and have revised the manuscript to ensure that “gender expression” terminology is used where relevant throughout. 

Title: The title needs to clarify that this refers to gender expression and not gender.

AUTHOR RESPONSE: We have revised the title to “Exploring differences between gender expressions in exercise self-efficacy and outcome expectations for exercise in individuals with stroke”.

Introduction: There needs to be better clarification of what gender expression is and how it differs from biological sex and gender identification e.g. I may be born female, identify as female but have a male gender expression. This differentiation was not clear to me, as a novice on gender expression, until I read the section on the BSRI-12.

AUTHOR RESPONSE: We thank the reviewer for this comment and agree it is important to clarify that sex and gender constructs are interrelated but not interchangeable. On lines page 3, lines 87-96, we have added definition that is specific to gender expression, and a definition of biological sex to reflect the differentiation between the constructs. 

Page 3, Line 87-96: Gender expression refers to the traits, thoughts and beliefs an individual possesses regarding their gender self-concept, whereby gender expression is most commonly expressed on a fluid spectrum of masculinity to femininity [20]. Importantly, an individual’s gender expression may differ from their biological sex at birth [18]. Sex is a biological construct, whereby an individual is characterized as being male or female according to genetics, anatomy, and physiology [18], which influence biological processes such as ageing and prevalence, diagnosis, severity, and outcomes of disease [21]. 

Although paragraph 3 introduces gender expression a more detailed description of gender expression and its constructs is warranted.

AUTHOR RESPONSE: On Page 3, Lines 87-91, we have added a detailed definition of gender expression. 

Page 3, Lines 87-91: Gender expression refers to the traits, thoughts and beliefs an individual possesses regarding their gender self-concept, whereby gender expression is most commonly expressed on a fluid spectrum of masculinity to femininity [20].

Paragraph 4 of the introduction refers to gender differences in exercise and outcome. Please clarify if this refers to biological sex, self-identified gender or gender expression.

AUTHOR RESPONSE: We thank the reviewer for this comment. As we refer to men and women and behavioral factors related to exercise in paragraph 4, we have added the specification around “gender expression”.

Page 4, Lines 102-105: An individual’s gender expression may have an important influence on their exercise self-efficacy and outcome expectations for exercise, whereby differences in exercise behaviours between individuals possessing masculine versus feminine gender expressions are reported early in life and persist into adulthood [23–31].

Results: It would be of interest to see what masculine expressions are related to exercise self-efficacy and outcome expectation. Would the authors consider analysing these 12 criteria separately. This may change the recommendation for clinical practice to not test for gender expression as a whole but those specific criteria. Your discussion does also point to other research which has looked at these individual characteristics.

AUTHOR RESPONSE: We thank the reviewer for this comment. The intent of the BSRI-12 tool is to classify participants as possessing masculine, feminine, androgynous or undifferentiated gender expressions based on the overall mean values obtained on the masculine and feminine subscales, in relation to the sample medians, respectively. 

As such, it is not advised to conduct analyses of individual items within each feminine and/or masculine subscale. Furthermore, as the constructs of gender (gender expression, roles, identity, etc.) are fluid, the developers of the BSRI would not advise relying on potentially one gender trait in isolation in clinical practice, and would rather advise the use of a holistic composite measure of gender expression, as we have presented. 

In our Discussion, we present strategies that are in line with competitive and dominant traits, as well as the perceptions and needs of others. These are overarching traits used to describe the items within each of the masculine and feminine subscales of the BSRI-12. For example, within the feminine subscale, “sympathetic”, “sensitive to needs of others” and “affectionate” can all be grouped under “perceptions and needs of others”, and thus reiterating that feminine gender expression encompasses a variety of traits. 

Discussion: Please take care to be consistent in your reference to gender and gender expression.

AUTHOR RESPONSE: We greatly appreciate the reviewer’s attention to detail. We have thoroughly reviewed the manuscript and have ensured the reference to gender expression is included as appropriate throughout. 

I would also question whether left/right dominance and the side affected by the stroke may affect once perception of self-efficacy and exercise outcome. I would suggest that this is explored in the analysis or discussed as a consideration for future research/limitation.

AUTHOR RESPONSE: The reviewer raises an intriguing suggestion of influence of hand dominance and side of stroke. There have been no previous reports of the association between these factors and psychosocial factors for exercise post-stroke, thus we did not opt to include these as variables in our analyses. Given our sample size, the covariates we selected were based on strong evidence of associations with gender expression and psychosocial outcomes for exercise in stroke.

Please check spelling of self-efficacy throughout as it is variable - hypenated, one word, two words.

AUTHOR RESPONSE: We have thoroughly reviewed the manuscript and have ensured adequate spelling of ‘self-efficacy’ throughout the manuscript. 

Reviewer #2: 

Introduction: The authors presented the introduction in a clear and structured manner. However, while the authors stated that there is a gap in knowledge about whether gender-related associations exist in the context of psychosocial factors for exercise, I feel that they need to expand a bit more on this in order to better explain the justification for the study and the study's original contribution to knowledge. The authors described the literature on stroke and exercise, and exercise and gender expression. I'm curious as to whether there is any literature about any gender-based factors in persons with stroke outside of exercise, which may add to the understanding of how gender-based factors may affect exercise in this population. This may help explain the 'gap' in the literature a bit better.

AUTHOR RESPONSE: We thank the reviewer for this comment. We acknowledge that although it is an evolving and greatly important area of research, there is very little literature around gender-based considerations in the stroke population. To date, research in this population has eluded to gender roles (representing the behavioral norms applied to men and women in society) (Tannenbaum C, Greaves L, Graham I, 2016; BMC Medical Research Methodology) commonly possessed by women are barriers to participation to stroke rehabilitation. For example, women with stroke are more likely to possess a primary caregiver role which limits their ability to prioritize their own needs, and less likely to view stroke as a major health concern thus limiting their motivation to participate in stroke rehabilitation or clinical trials (Carcel & Reeves 2021; Stroke). We have added these insights on page 3 (Line 79-91) to reinforce the need for the current study. 

Page 3, Lines 79-91: There have been preliminary reports of gender-based considerations being associated with participation in stroke rehabilitation programs, specifically around barriers faced by women. With regards to the construct of gender roles (representing the behavioral norms applied to men and women in society) [18], women with stroke are more likely to possess primary caregiver roles, whereby it is increasingly difficult to focus on their own health when pressures to prioritize the needs of others are present [19]. Moreover, women are less likely to view stroke as a major health concern which limits their motivation to participate in stroke rehabilitation [19]. These gender roles may also influence exercise behaviours. 

Gender expression is a construct of gender that has not been studied in individuals with stroke but may be an important factor associated with exercise self-efficacy and outcome expectations for exercise. Gender expression refers to the traits, thoughts and beliefs an individual possesses regarding their gender self-concept, whereby gender expression is most commonly expressed on a fluid spectrum of masculinity to femininity [20].

We do want to emphasize that our study is the first to date to explore the construct of gender expression and how it relates to exercise-related behaviours after stroke. 

Analysis/Methods: I have concerns about the sample size. There is no indication of a power analysis or sample size estimation calculations described so it is difficult to say whether the study was sufficiently powered.

AUTHOR RESPONSE: We thank the reviewer for this comment. There have been no previous studies examining gender-based considerations in individuals with stroke, thus we did not have previously reported group means and variance/standard deviations for a rigorous sample size calculation. We have positioned this study as an exploratory study for which we hope future research that may benefit from our data to inform ANCOVA sample size calculations. We also emphasize the balanced cells between BSRI-12 groups as a substantial strength of our study. Nonetheless, we have added the lack of sample size calculation as a limitation of the study (Page 19, Lines 395-398). 

Page 19 (Lines 395-398): Moreover, given the exploratory nature of this study and sparsity of previous literature on gender-based based considerations in this population, we did not perform a sample size calculation. Our data may inform future research to consult our group means and variances.

Also, while the sample was one of convenience from community, and databases, how did they go about recruiting from these? What was their recruitment process - phone, flyers - how were participants contacted? How did they assess for the inclusion/exclusion criteria?

AUTHOR RESPONSE: On page 6, 156-160, we elaborate on our recruitment strategies. 

Page 6, Line 156-160: Members of the research team attended monthly meetings of community stroke groups and provided a brief presentation of the research study, distributed study flyers, addressed questions, and collected phone numbers of interested individuals. For all recruitment methods, participants were screened for eligibility for the study over the phone with a member of the study team. 

There were no tests performed on the independent variables to determine whether the groups were different in any of the demographic factors. Also, the gender groups and demographic characteristics of the outliers should be mentioned.

AUTHOR RESPONSE: We agree with this comment. We have revised Table 1 to also include disaggregated data for each gender expression group. On page 14 (lines 297-300) we provided a summary of the characteristics of the three outliers. In order to provide as much information possible into the characteristics of the three outliers, we have uploaded a supplementary file that presents all their demographic information.

Page 14, Lines 297-300: The outliers represent three different gender expression groups, ranged in age and predominantly experienced mild to moderate ischemic strokes. A detailed description of the characteristics of three outliers are provided in supplementary file 1.

Is the median-split a valid way to categorise participants into gender groups? The authors gave no justification for doing the classification this way. How does the BSRI-12 classify persons? Why did the authors not use this method of classification? I did a brief search of the tool and found that the median-split has been used as a way to classify participants into gender roles. This should be better detailed in the study.

AUTHOR RESPONSE: We thank the reviewer for this comment and we would like to clarify our selection of this scoring method. Firstly, the seminal papers by Bem 1981 and Spence 1975 cited on page 8 (Line 184) addresses the strengths of using the median-split method for scoring the BSRI. Secondly, the median split method is commonly used for samples with older adults or chronic conditions, as the alternative normative value approach established by Sandra Bem was conducted in younger youth populations and would not be reflective of our population. We have added a statement in on Page 8 (Lines 184-187) to explain our choice of the median-split method.

Page 8, Lines 184-187: We used the median-split to classify the BSRI-12 [49,50]. This commonly used approach in older adult populations [51,52] was selected as the alternative approach to using normative values which was based on a sample of younger adults [22], and thus not representative of our sample. 

We would like to clarify that the terminology “gender roles” and “gender expression” is used interchangeably in the literature, though we strongly argue that gender roles is not appropriate to describe the specific construct measured by the BSRI-12. Gender roles “represent the behavioral norms applied to men and women in society, which influence individuals’ everyday actions, expectations, and experiences (e.g., caregiver and housework responsibilities)” (Tannenbaum C, Greaves L, Graham I, 2016, BMC Medical Research Methodology), whereas gender expression represents the traits that individuals use to describe their behaviours and personality (Beltz 2021; Sci Rep). Indeed, the latter definition is reflective of the constructs assessed by the BSRI-12.

Overall:

There are a few grammatical errors but these do not affect the understanding of the article and can be easily rectified.

The suggestions for future research are very good given the cited limitations of the study.

Overall, the study is relevant to today's evolving society and highlights important considerations, however, there are some recommended corrections that I think would be helpful in improving the validity of the study.

AUTHOR RESPONSE: We appreciate the positive feedback and constructive criticism from Reviewer #2 and have incorporate the feedback to help improve the manuscript. 

Reviewer #3: I enjoyed reviewing this paper. The topic is quite interesting and relevant for tx planning in stroke rehab. The paper is generally solid in terms of clarity and style of writing. The analysis is articulated clearly.

I have indicated "major" revision somewhat arbitrarily, mainly because I have several comments that I think would strengthen the paper, but I think you will be readily able to address these.

AUTHOR RESPONSE: Thank you to Reviewer #3 for the positive feedback and for the tremendously helpful feedback, of which we hope we adequately addressed in the revised manuscript.

First, a few straightforward editing recommendations:

Line 96: Word choice. Is "transpire" what you mean. Consider "persist" or "continue."

97: Should read: "...people with masculine gender traits are more likely..."

99: "reported" should read "report" (be consistent with tense)

116: should be "knowledge of" or "knowledge regarding"

124:The use of "outcomes" without qualification here is a bit confusing. Would be better to specify that you expect lower self-efficacy and expectations so the reader doesn't conflate scale scores with exercise outcomes.

137: "ethic" should be "ethics"

182-83: I suggest eliminating the text in parentheses.

300: "between" should be "among"

In general, you should be consistent with use of Oxford commas.

AUTHOR RESPONSE: We greatly appreciate the editing feedback provided by Reviewer #3. We have made all the revisions accordingly.

Now, a few more substantive recommendations:

1. When you first mention outliers, the number is not indicated. Later, you mention removing outliers in the analysis of the SOEE. I think you provide a clear statement about all outliers removed and when. This is especially important given your relatively small sample size.

AUTHOR RESPONSE: Three outliers were removed for the SOEE analysis. When deciding whether outliers should be removed, we conducted visual inspection (Q-Q plots) and formal analyses (e.g., inspected Cook’s d values, influence) to ensure that these values were truly outliers. We found that assumptions for homogeneity and distribution to run ANOVA and ANCOVA were violated. We acknowledge the relatively small sample size, but the removal of these 3 outliers were critical to ensure that assumptions for running our statistical analyses were met. On page 10 (lines 238-242), of the original manuscript, we provided insight into the statistical analyses conduct to inspect for clear outliers. 

On Page 10, Line 248, we added wording to clarify that three outliers were removed for our SOEE models:

Page 10, Line 241: In the SOEE models, three outliers with high residuals and influence were removed and models were re-evaluated.

On page 14 (lines 297-300), we summarized characteristics of the three outliers. We have also uploaded a supplementary file that provides the characteristics of the three outliers from the SOEE model.

Page 14, Lines 297-300: The outliers represent three different gender expression groups, ranged in age and predominantly experienced mild to moderate ischemic strokes. A detailed description of the characteristics of three outliers are provided in supplementary file 1.

2. I see that you reported alphas for the BSRI subscales, but I don't see them for the other instruments. Reporting internal consistency for all scales is a good idea. Also, you might consider reporting McDonald's Omega, given criticisms of the restrictive assumptions of alpha. I recently had this critique myself from a reviewer and just reported both.

AUTHOR RESPONSE: We greatly appreciate the reviewer sharing insight to their own experiences with reporting internal consistency. We computed internal consistency values for all scales and have reported both alpha and McDonald’s Omega values for the BSRI-12, SEPA and SOEE.

Page 10, Lines 230-231: Cronbach’s alpha and McDonalds’ omega were computed to determine internal consistency of the SOEE and SSEE in our sample.

Page 11, Lines 261-263: The masculine (Cronbach’s alpha 0.86, McDonald’s Omega 0.87) and feminine subscales (Cronbach’s alpha 0.89, McDonald’s Omega 0.89) of our sample had high internal consistency.

Page 12, Line 278-279: The SEPA demonstrated good internal consistency in our sample (Cronbach’s alpha 0.75, McDonald’s omega 0.76).

Page 14, Lines 293-294: The SOEE demonstrated high internal consistency in our sample (Cronbach’s alpha 0.86, McDonald’s omega 0.87).

3. You mention some literature on people with undifferentiated traits, but I think this deserves more elaboration in your discussion. You expected to find other differences as well between masculine and feminine traits, so the finding stands out. More discussion on personality correlates and health outcomes for people with undifferentiated traits would flesh out the paper--and be relevant for clinicians working with this group.

AUTHOR RESPONSE: We thank the reviewer for this comment on elaborating on the literature on individuals possessing undifferentiated gender traits. This is a severely understudied area of research, but we have added more discussion around these individuals on page 16 (lines 329-340).

Page 16, Lines 329-340: Moreover, individuals with undifferentiated gender traits are more likely to report greater levels of mobility disability [51], social anxiety [69] and depression [70] in contrast to other gender expressions. Indeed, low self-efficacy is often a precursor for social anxiety [71]. The current study adds to the growing body of evidence describing poorer psychosocial health outcomes of individuals with undifferentiated gender traits, and highlights the need to develop strategies to promote exercise self-efficacy specific to the needs of these individuals. Such strategies may include greater emphasis placed on individualized exercise programs with clinicians working 1:1 with patients, or involving other patients with similar functional abilities since individuals with undifferentiated gender expressions may not possess competitive nor dominant traits, nor focus on the perceptions and needs of others [22,42]. These strategies may not only promote exercise self-efficacy but may also minimize social anxiety.

4. Your sample size mey have limited power to find some differences. Addressing this explicitly in limitations and recommendations would add to the paper.

AUTHOR RESPONSE: We thank the reviewer for this comment. There have been no previous studies examining gender-based considerations in individuals with stroke, thus we did not have previously reported group means and variance/standard deviations for a rigorous sample size calculation. We have positioned this study as an exploratory study for which we hope future research that may benefit from our data to inform ANCOVA sample size calculations. We also emphasize the balanced cells between BSRI-12 groups as a substantial strength of our study. Nonetheless, we have added the lack of sample size calculation as a limitation of the study (Page 19, Lines 395-398). 

Page 19 (Lines 395-398): Moreover, given the exploratory nature of this study and sparsity of previous literature on gender-based based considerations in this population, we did not perform a sample size calculation. Our data may inform future research to consult our group means and variances.

---

## [Editor Report · Decision Letter 1]

8 Feb 2024

Exploring differences between gender expressions in exercise self-efficacy and outcome expectations for exercise in individuals with stroke

PONE-D-23-30818R1

Dear Dr. Tang,

We are pleased to inform you that your manuscript has been judged scientifically suitable for publication and will be formally accepted for publication once it meets all outstanding technical requirements.

Kind regards,

Henry Hugh Bailey, Ph.D

Academic Editor

PLOS ONE

---

## [Editor Report · Acceptance letter]

4 Mar 2024

PONE-D-23-30818R1 

PLOS ONE

Dear Dr. Tang, 

I'm pleased to inform you that your manuscript has been deemed suitable for publication in PLOS ONE. Congratulations! Your manuscript is now being handed over to our production team.

Kind regards, 

on behalf of

Dr. Henry Hugh Bailey 

Academic Editor

PLOS ONE